# SnS_2_ Nanosheets with RGO Modification as High-Performance Anode Materials for Na-Ion and K-Ion Batteries

**DOI:** 10.3390/nano11081932

**Published:** 2021-07-27

**Authors:** Leqiang Wu, Hengjia Shao, Chen Yang, Xiangmin Feng, Linxuan Han, Yanli Zhou, Wei Du, Xueqin Sun, Zhijun Xu, Xiaoyu Zhang, Fuyi Jiang, Caifu Dong

**Affiliations:** School of Environmental and Material Engineering, Yantai University, Yantai 264005, China; wlq_246@163.com (L.W.); shao0331@163.com (H.S.); yc083625@163.com (C.Y.); fxm18763285297@163.com (X.F.); a13156280283@163.com (L.H.); zhouyanli@ytu.edu.cn (Y.Z.); duwei@ytu.edu.cn (W.D.); sxq@ytu.edu.cn (X.S.); xuzhijun@ytu.edu.cn (Z.X.); zhangxiaoyu@ytu.edu.cn (X.Z.)

**Keywords:** SnS_2_, anode, sodium-ion batteries, potassium-ion batteries

## Abstract

To date, the fabrication of advanced anode materials that can accommodate both Na^+^ and K^+^ storage is still very challenging. Herein, we developed a facile solvothermal and subsequent annealing process to synthesize SnS_2_/RGO composite, in which SnS_2_ nanosheets are bonded on RGO, and investigated their potential as anodes for Na^+^ and K^+^ storage. When used as an anode in SIBs, the as-prepared SnS_2_/RGO displays preeminent performance (581 mAh g^−1^ at 0.5 A g^−1^ after 80 cycles), which is a significant improvement compared with pure SnS_2_. More encouragingly, SnS_2_/RGO also exhibits good cycling stability (130 mAh g^−1^ at 0.3 A g^−1^ after 300 cycles) and excellent rate capability (520.8 mAh g^−1^ at 0.05 A g^−1^ and 281.4 mAh g^−1^ at 0.5 A g^−1^) when used as anode for PIBs. The well-engineered structure not only guarantees the fast electrode reaction kinetics, but also ensures superior pseudocapacitance contribution during repeated cycles, which has been proved by kinetic analysis.

## 1. Introduction

With the intensification of environmental pollution, green renewable energy has currently become an active area of research [1,2]. Among numerous energy storage devices, sodium-ion batteries (SIBs) and potassium-ion batteries (PIBs) have gained extensive concern for energy storage because of their similar energy storage mechanisms to lithium-ion batteries (LIBs) and the abundant sodium and potassium resources [3,4,5]. However, the low capacity of commercial graphite anodes for Na-ion and K-ion storage has greatly limited the large-scale development of SIBs and PIBs [6,7,8]. Therefore, it is imperative to develop high-performance anode materials for both SIBs and PIBs.

Specifically, the hexagonal tin (IV) sulfide (SnS_2_) possesses unique two-dimensional (2D) layered structure with large interlayer spacing and high specific capacity based on both conversion and alloying processes [9,10,11], which makes it more appropriate for Na-ion and K-ion storage. Unfortunately, similar to other TMSs, SnS_2_ also has the low intrinsic conductivity and the dramatic volume and structural changes, which will easily lead to the poor sluggish kinetics and rapid capacity reduction in the process of Na^+^ and K^+^ insertion/extraction [12,13,14,15]. The construction of SnS_2_ and various conductive carbonaceous composite has become one of the most effective approaches to ameliorate these problems [16,17,18]. For instance, the carbon-coated SnS_2_ nanosheet composite fabricated by Li et al., delivered a reversible capacity of ~420 mAh g^−1^ at a current density of 500 mA g^−1^ after 100 cycles for Na^+^ storage [19]. SnS_2_ and rGO composite synthesized by Glushenkov, displayed a reversible capacity of 250 mAh g^−1^ at 25 mA g^−1^ after 30 cycles for PIBs [20]. Even though the cycling stability has been improved via carbon modified anode materials, how to achieve higher-performance SnS_2_ electrode with long cycle life is still a huge challenge for SnS_2_.

Herein, SnS_2_/RGO nanoarrays have been fabricated via a facile solvothermal method along with a subsequent annealing process. GO is chosen as the substrate because the rich functional groups on GO can increase the nucleation sites with metal ions, which contributes to form a strong and close-coupled interface between RGO and SnS_2_. Benefiting from the unique structural characteristics, SnS_2_/RGO displays higher cycle stability and better rate capability than the pristine SnS_2_. For example, the prepared SnS_2_/RGO shows a high reversible capacity of 653.8 mAh g^−1^ at 100 mA g^−1^, and outstanding rate performance (593 mAh g^−1^ at 200 mA g^−1^ and 400 mAh g^−1^ at 2000 mA g^−1^) for SIBs. Moreover, SnS_2_/RGO also displays a high-rate capacity (520.8 mAh g^−1^ at 50 mA g^−1^ and 281.4 mAh g^−1^ at 500 mA g^−1^) and good cycling stability (130 mAh g^−1^ at 300 mA g^−1^ after 300 cycles) for K-ion storage. The fast electrode reaction kinetics and superior pseudocapacitance contribution may be the reason for the excellent performance, which has been proved by kinetic analysis. Thus, the present study provides a novel strategy to prepare SnS_2_ and rGO composite with excellent performance for both sodium and potassium storage and this strategy which may be extended to fabricate other high-performance electrode materials for energy storage.

## 2. Materials and Methods

### 2.1. Synthesis of SnS_2,_ and SnS_2_/RGO Nanoarrays

In a typical preparation, 10 mg GO (prepared by the modified Hummers method) was first dispersed in 8 mL isopropanol. Then 0.172 mmol (38.8 mg) SnCl_2_∙2H_2_O and 0.761 mmol (62.4 mg) 2-methylimidazole (2-MIN) were added into the above solution and stirred vigorously for 12 h at 25 °C. After that, 0.8 mmol (60 mg) thioacetamide (TAA) was added into the solution under stirring. After stirring for another 2 h, the solution was transferred into a Teflon-lined autoclave (25 mL) and kept at 120 °C for 24 h. The solid product was collected and dried. Finally, the precursor annealed at 350 °C with a heating rate of 2 °C min^−1^ in Ar gas for 1 h to gain the final products of SnS_2_/RGO. The SnS_2_ was produced by similar route, but with no added organic molecules and GO in the solvothermal process.

### 2.2. Materials Characterization

The morphologies of SnS_2_/RGO and SnS_2_ were tested by a transmission electron microscope (JEOL-1400 Plus, Tokyo, Japan), the high-resolution TEM (HRTEM JEOL-2011, Tokyo, Japan) and a field emission scanning electron microscope (FESEM, ZEISS Geminisem 300, Oberkochen, Germany). The components of SnS_2_/RGO and SnS_2_ were analyzed by energy dispersive spectrometry (EDS). Crystallographic phases of SnS_2_/RGO and SnS_2_ were measured by powder X-ray diffraction (XRD Bruker, D_8_-Advanced, Tokyo, Japan) using a Cu Kα radiation. Raman spectroscopy (LabRAM HR 800, Paris, France) with 532 nm laser excitation. X-ray photoelectron spectroscopic (XPS) surveys were performed on an X-ray photoelectron spectrometer (Thermo Scientific Escalab 250Xi, New York, NY, USA), which uses an Al Kα as the excitation source.

### 2.3. Electrochemical Measurements

For the Na-ion half cells, 2032 coin-type cells were employed. The electrodes were obtained by applying a slurry mixture consisting of active materials (70 wt.%), Super-P (20 wt.%), and polyvinylidene difluoride (PVDF, 10 wt.%), to a copper foil and drying it under vacuum at 60 °C for 6 h. The loading of the active material on each disc was about 1.05–1.4 mg cm^−2^. A sodium foil with a diameter of about 14 mm was prepared in a glove box under the protection of high-purity argon as a counter electrode using sodium block (Aladdin, 99.7%). The glass fiber (Whatman) was used as the separator and 1 M NaPF_6_ dissolved in the diethyleneglycoldimethylether acted as electrolyte. To study the electrode performance of SnS_2_/RGO for PIBs, CR 2016 coin batteries was built using SnS_2_/RGO as anode, potassium metal was used as counter electrodes. The electrolyte was 3.0 M potassium bis(fluorosulfonyl)imide (KFSI) in TGM. The anode was prepared by casting slurries of active material, super P and PVDF binder in a mass proportion of 7:2:1 onto copper foil with an active material loading of around 1.05–1.4 mg cm^−2^. The galvanostatic charge/discharge tests were conducted on battery test station (LAND CT-2001A, Wuhan, China) from 0.01–3.0 V. Cyclic voltammetry (CV) tests (0.01–3.0 V) were carried on the CHI 760E electrochemical workstation.

## 3. Results and Discussion

Figure 1 schematically illustrates the fabrication route of SnS_2_/RGO and SnS_2_. The Sn-based nanosheets in-situ grown on graphene oxide (GO) substrate can be obtained via solvothermal reaction by added 2-methylimidazole (2-MIN) and GO substrate in the first step. In the absence of 2-MIN and GO substrate, flower-like three-dimensional microspheres consisting of larger nanosheets are obtained. After a calcination process under Ar atmosphere, SnS_2_/RGO and SnS_2_ are obtained.

The FESEM and TEM are carried out to investigate the morphological features of as-prepared materials. Figure 2a and Appendix A show the detailed morphological characteristics of the precursors. As seen in Figure 2a, the vertical tin-based precursor nanosheets are densely and uniformly grown on both sides of the GO substrate, forming a sandwich-like hierarchical structure. To find out the function of 2-MIN, we performed several comparative experiments by varying the reaction conditions (Appendix A). In the absence of 2-MIN, the same synthesis process produced bare GO sheets and parts of the nanosheets are assembled separately to form a flower-like structure (Appendix A). In the presence of 2-MIN but without GO, the results showed that the nanosheets are grown on the solid spheres (Appendix A). In the absence of 2-MIN and GO substrates, the Sn-based precursor shows flower-like three-dimensional spherical structure assembled by larger nanosheets compared with those in the presence of 2-MIN (Appendix A). The results show that 2-MIN can control the size of precursor nanosheets and induce the growth of nanosheets on GO substrates, which helps to synthesize some novel nanomaterials. FESEM images at different magnifications (Figure 2b,c) show that the nanosheet structure is well retained after annealing at 350 °C. TEM image in Figure 2d further indicates that SnS_2_ nanosheet is tightly grown on microns GO nanosheets, indicating the synthesized sandwich structure integrates the features of micro- and nanostructures. Figure 2e displays a typical high-resolution transmission electron microscopy (HR-TEM) image of SnS_2_/RGO, the d-spacing of 0.338 nm corresponding to the (100) plane in SnS_2_. Moreover, the selected-area electron diffraction (SAED) pattern (Figure 2f) shows the tagged diffraction rings can be well indexed to (100), (101), (110) and (111) crystal planes of SnS_2_, respectively. The FESEM image and the corresponding Energy dispersive X-ray spectroscopy (EDX) elemental mapping images of SnS_2_/RGO are shown in Figure 2g, revealing the homogeneous dispersion of Sn, S, and C throughout the composites, further verifying the nanosheets are well dispersed on the GO substrate throughout the whole network. As shown in Appendix A, pure Sn-based precursor can also keep the flower structure after annealing. Moreover, the EDS mapping result shown in Appendix A displays the homogeneous distribution of Sn and S elements throughout the flower-like SnS_2_ sphere.

The composition of two samples is investigated by X-ray diffraction (XRD). It can be seen that SnS_2_/RGO and SnS_2_ show similar diffraction peaks (Figure 3a) and the diffraction peaks can be attributed to hexagonal SnS_2_ with P-3m 1 (164) space group (JCPDS no. 23-0677), corresponding to the above mentioned HRTEM and SAED results. There are no obvious diffraction peaks for the carbon phase due to its poor crystallinity. Figure 3b shows a typical 2D layered crystal structure of SnS_2_, with an interlayer spacing of 0.59 nm. The Raman spectra of SnS_2_/RGO and GO are shown in Figure 3c. It can be seen that SnS_2_/RGO and GO have distinct characteristic peaks at ~1344 and 1602 cm^−1^, which can be attributed to the D (disordered carbon) and G (graphitic carbon) bands of graphite. Moreover, the intensity ratio of D-band to G-band (I_D_/I_G_) is 1.06, 1.37 for GO, and SnS_2_/RGO, respectively, indicating more defects in SnS_2_/RGO [21,22]. In addition, SnS_2_/RGO exhibits a peak located at 313 cm^−^^1^, which corresponds to the A_1g_ vibration of SnS_2_ [23,24]. Appendix A presents the N_2_ adsorption-desorption isotherm profiling of SnS_2_/RGO. The relatively large surface area (33.885 m^2^ g^−1^) and abundant pores (0.159 cm^3^ g^−1^) of SnS_2_/RGO can provide adequate active sites and promotes the rapid transport of electron and ions. X-ray photoelectron spectroscopy (XPS) was further carried out to investigate the surface electronic states and the chemical compositions of SnS_2_/RGO. The characteristic peaks of Sn 3d, S 2p, C 1s, and O 1s are observed in the wide survey spectrum (Appendix A). Figure 3d shows the Sn 3d spectrum, in which two strong peaks at 486.1 and 494.5 eV correspond to Sn 3d_5/2_ and Sn 3d_3/2_ of Sn^4+^ in SnS_2_/RGO [25,26]. The high-resolution S spectrum can be convoluted into two peaks at 162.4 eV and 161.3 eV, corresponding to S 2p_3/2_ and S 2p_1/2_ of S^2−^ (Figure 3e), which further confirms the formation of SnS_2_ [27,28]. As presented in Figure 3f, the C 1s spectrum can be divided into two peaks and assigned to the C-C (284.8 eV) and C-O (286.5 eV) bonds, respectively [29,30].

Inspired by their special structure, the electrochemical performance of the obtained SnS_2_/RGO and SnS_2_ for SIBs anodes was tested. Figure 4a,b display the galvanostatic charge-discharge (GCD) curves of SnS_2_/RGO and SnS_2_ at 0.1 A g^−1^, respectively. It can be clearly seen that SnS_2_/RGO reveals the better performance than pure SnS_2_. For SnS_2_/RGO electrode, the discharge and charge capacities of the first cycle are 795.4 and 653.8 mAh g^−1^, respectively. The initial loss of irreversible capacity is mainly attributable to the irreversible process of electrolyte decomposition and the generation of solid-electrolyte interface (SEI) layers on the electrode surface, which was a common phenomenon in tin-based sulfide anode materials [31,32]. After the initial cycle, the GCD curves are almost repeated in the flowing cycles, which is coinciding with the CV results (Appendix A), indicating the good stability of SnS_2_/RGO electrode. For further comparison, the cycling performance of SnS_2_/RGO and SnS_2_ was measured at 0.5 A g^−1^ and shown in Figure 4c. SnS_2_/RGO displays the better performance than pure SnS_2_ electrode and maintains a high capacity of 581 mAh g^−1^ after 80 cycles. The SnS_2_/RGO exhibits excellent sodium storage performance that exceeds many previously reported anode materials for SIBs (Appendix A). Beside the cycling performance, the SnS_2_/RGO electrode also showed excellent rate performance (Figure 4d). It can show specific capacities of 593, 490.5, 425.2, 431.8, 428.1 and 400 mAh g^−1^ at 0.2, 0.4, 0.6, 0.8, 1 and 2 A g^−1^, respectively. When the current was switched back to 0.2 A g^−1^, the specific capacity of 491.3 mAh g^−1^ can be regained, which indicated that the SnS_2_/RGO electrode has good structural stability. To verify the excellent rate performance of SnS_2_/RGO, the morphology of SnS_2_/RGO electrode for SIBs after rate performance was observed by FEEM. It can be seen from Appendix A that the SnS_2_ nanosheets become nanoparticles, but still in close contact with graphene, which is an essential aspect for the good rate performance of the material. To further understand the improved electrochemical performance of SnS_2_/RGO electrode, the EIS spectra of the SnS_2_/RGO and SnS_2_ electrodes are measured before and after 10 cycles. As shown in Figure 4e, SnS_2_/RGO shows a much smaller charge transfer impedance (Rct) than pure SnS_2_ after 10 cycles at 0.2 A g^−1^, indicating that the improvement of conductivity benefiting from the introduction of RGO. Moreover, the Warburg coefficient of SnS_2_/RGO is 46.6 Ω s^−^^0.5^ (Figure 4f), which is much smaller than that of SnS_2_ (316.1 Ω s^−^^0.5^), showing that Na-ion has a faster diffusion ability in SnS_2_/RGO. These results are coincident with their sodium storage performance.

To better understand the superior capability of SnS_2_/RGO, the charge storage behavior and reaction kinetics of SnS_2_/RGO and SnS_2_ are further analyzed according to CV and GITT tests. Figure 5a and Appendix A show the CV curves of SnS_2_/RGO and SnS_2_ at multiple scan rates from 0.1 to 0.6 mV s^−1^. It should be pointed out that the curves keep its shape even at a high scan rate of 0.6 mV s^−^^1^. The dependency between the peak current (i) and the sweep rate (ν) is based on Equation (1): i = αν^b^, (1)
where b reflects the charge storage behavior [33,34,35]. Figure 5b and Appendix A show the b-values of the two redox peaks for SnS_2_/RGO and SnS_2_, respectively. It can be seen that SnS_2_/RGO electrode shows a larger b-values than SnS_2_ electrode, indicating the pseudocapacitive contribution ratio of SnS_2_/RGO is larger than that of the SnS_2_ electrode. The ratios of capacitive contribution can be evaluated by Equation (2) [36,37,38]:(2)$$i(v)=k_{1}v+k_{2}v^{1/2}$$
where the *k*_1(*V*)_*v* and the *k*_2(*V*)_*v*^1/2^ stand for the capacitive-controlled contribution and the diffusion-controlled contribution, respectively. Figure 5c and Appendix A show the typical CV profiles of the Na-ion capacitive (dark red region) in comparison with the total measured current for SnS_2_/RGO and SnS_2_ at the scan speed of 0.6 mV s^−1^. The capacitive-controlled contribution is calculated to be 83.16% of the total Na^+^ storage at 0.6 mV s^−1^ in SnS_2_/RGO, which is higher than that in the case of SnS_2_ (74.13%). In addition, the ratios of pseudocapacitive contribution are all increased as the scan rate increases but from 62.76 to 83.16% for SnS_2_/RGO, 44.56 to 74.13% for SnS_2_ (Figure 5d). Therefore, the improvement of the rate capability may be related to the enhanced contribution ratios of the capacitive behaviors. After that, the galvanostatic intermittent titration technique (GITT) was carried out to further evaluate the Na^+^ solid-state diffusion dynamics of SnS_2_/RGO and SnS_2_ [39]. The GITT test was performed on at 0.1 A g^−1^ in a voltage range of 0.01–3 V (Figure 5e). Appendix A shows the detailed test and calculation method. As shown in Figure 5f–i, SnS_2_/RGO shows lower reaction resistance and higher D values during the entire cycle than that with SnS_2_, which can be probably responsible for the superb performance.

The electrochemical performance of SnS_2_/RGO as the anode material for PIBs was investigated using CR2016 half cells. Figure 6a exhibits the GCD profiles of SnS_2_/RGO at 50 mA g^−1^. The discharge/charge capacity of the first cycle is 983.1/520.8 mAh g^−1^, respectively. The irreversible capacity is considered to be the formation of the SEI layer, which is a common phenomenon in transition metal-based anode materials [40]. In the following cycles, the profiles are well overlapped, which is coinciding with the CV results (Appendix A). In addition, it shows a high capacity of 520 mAh g^−1^ after 10 cycles. Figure 6b exhibits the rate performance of SnS_2_/RGO. The reversible capacities of the SnS_2_/RGO are 520.8, 405, 337.5, and 336.7 mAh g^−1^ mAh g^−1^ at 50, 100, 200 and 300 mA g^−1^, respectively. Even at 500 mA g^−1^, it still can remain a high capacity of 281.4 mAh g^−1^, which indicates the excellent rate capability of SnS_2_/RGO for K-ion storage. To further evaluate the cyclic stability of SnS_2_/RGO electrode, the cycling test under current density of 100 and 300 mA g^−1^ are conducted. The SnS_2_/RGO shows a high reversible capacity of 403.2 mAh g^−1^ at 100 mA g^−1^ after 80 cycles, with coulombic efficiency of almost 100% (Figure 6c). The long cycle performance of SnS_2_/RGO electrode is shown in Figure 6d. It delivers a high reversible capacity of 130 mAh g^−1^ at 0.3 A g^−1^ after 300 cycles. The results have been amply vindicated that the SnS_2_/RGO electrode has excellent cycling stability for K-ion storage. To gain insight into the electrochemical kinetics of K-ion storage in SnS_2_/RGO, we performed CV (0.2 to 1.0 mV s^−1^) and GITT tests. As shown in Figure 6e–h and Appendix A, the large proportion of pseudocapacitive contribution, and high D values enables improvement of electrochemical performance, especially the rate capability. In addition, SnS_2_/RGO also displays the excellent Li-ion storage performance (Appendix A). All in all, this hybrid structure demonstrates good electrode integrity, fast electrode reaction kinetics and the superior cycling stability, making SnS_2_/RGO a great potential for the future electrochemical energy storage.

## 4. Conclusions

In summary, the well-designed sandwich-like hierarchical SnS_2_/RGO structure have been synthesized via a facile solvothermal and subsequent annealing process. We found that 2-MIN can induce the growth of SnS_2_ nanosheets on GO substrate during the solvothermal process, which has not been reported before. Benefiting from the good electrode integrity, excellent pseudocapacitive contribution and fast electrode reaction kinetics, SnS_2_/RGO displays an outstanding rate performance (593 mAh g^−1^ at 0.2 A g^−1^ and 400 mAh g^−1^ at 2.0 A g^−1^) and good cycling stability with a high capacity of 581 mAh g^−1^ at 0.5 A g^−1^ over 80 cycles when applied as anode for SIBs. Simultaneously, it delivers a high capacity of 130 mAh g^−1^ at 0.3 A g^−1^ after 300 cycles as anode for PIBs. These results indicate that the SnS_2_/RGO is a promising anode material for electrochemical energy storage.

## Figures and Tables

**Figure 1 nanomaterials-11-01932-f001:**
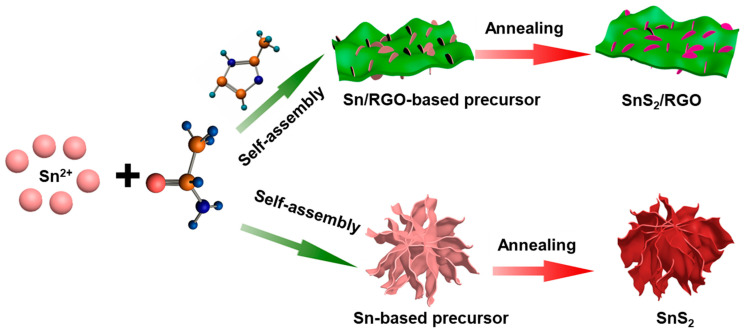
Schematic illustration of the formation process of pure SnS_2_, and SnS_2_/RGO.

**Figure 2 nanomaterials-11-01932-f002:**
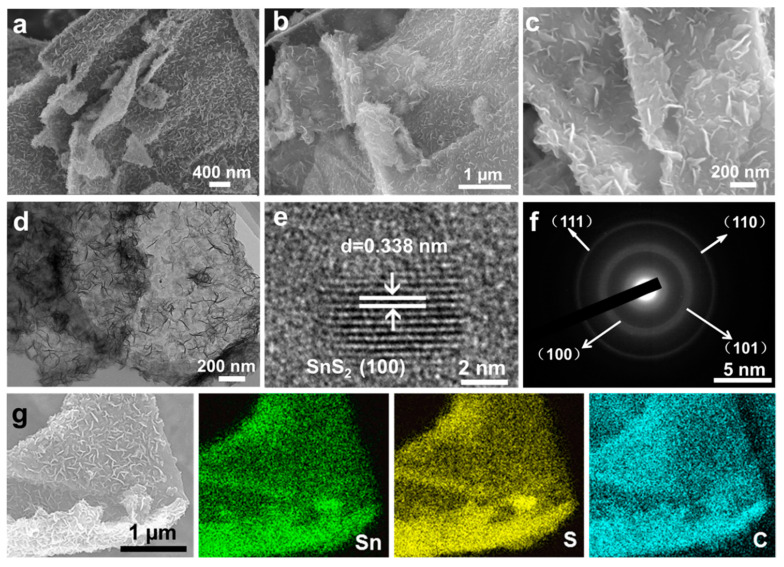
(**a**) FESEM image of Sn/RGO-based precursor. (**b**,**c**) FESEM images of SnS_2_/RGO at different magnifications. (**d**) TEM image of SnS_2_/RGO. (**e**) HRTEM image of SnS_2_/RGO. (**f**) Corresponding SAED pattern. (**g**) A typical FESEM image of SnS_2_/RGO and the corresponding elemental mappings of Sn, S and C elements (as labeled).

**Figure 3 nanomaterials-11-01932-f003:**
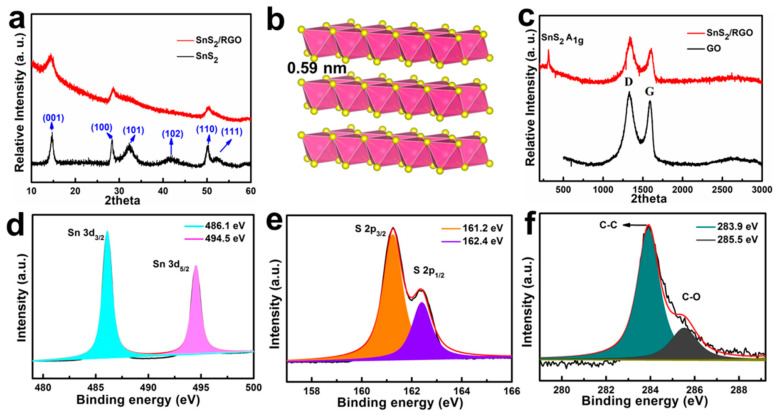
(**a**) XRD patterns of the obtained SnS_2_/RGO and SnS_2_. (**b**) Crystal structure of SnS_2_. (**c**) Raman spectra of SnS_2_/RGO and GO. The high-resolution XPS spectra of Sn 3d (**d**), S 2p (**e**), and (**f**) C 1s of SnS_2_/RGO.

**Figure 4 nanomaterials-11-01932-f004:**
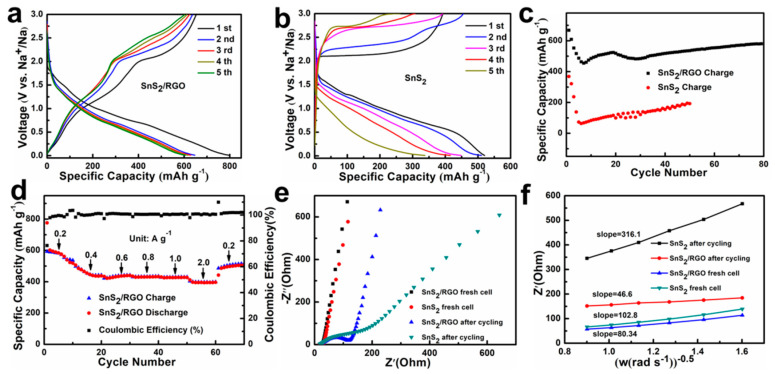
(**a**,**b**) Discharge/charge curves of SnS_2_/RGO and SnS_2_ at 0.1 A g^−1^. (**c**) The cycling performance of SnS_2_/RGO and SnS_2_ at 0.5 A g^−1^. (**d**) Rate capability at different current densities (0.2–2 A g^−1^). (**e**) The electrochemical impedance spectra of the SnS_2_/RGO and SnS_2_ were obtained before and after 10 cycles at 0.2 A g^−1^. (**f**) The relationship between the Z_re_ and ω^−1/2^ in the low frequency of SnS_2_/RGO and SnS_2_.

**Figure 5 nanomaterials-11-01932-f005:**
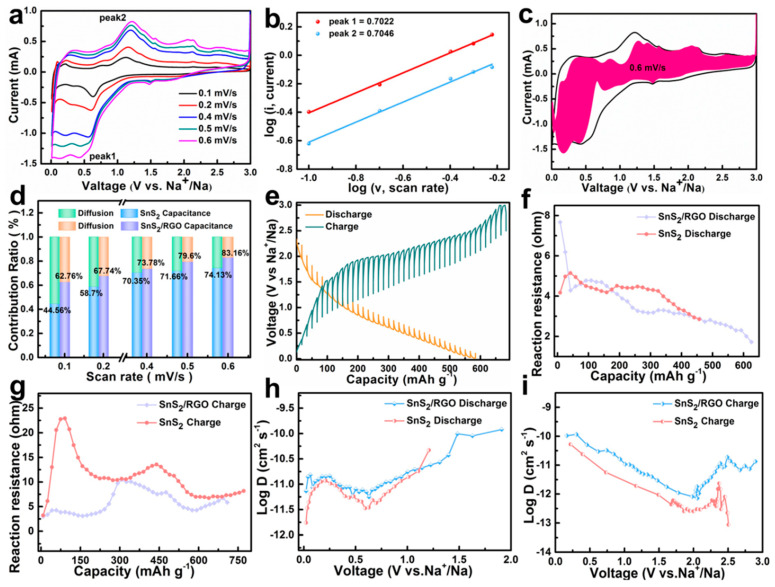
(**a**) CV profiles of SnS_2_/RGO at different sweep rates. (**b**) Linear relationship of log (**i**) vs. log (v) plots at each redox peak. (**c**) CV curve with capacitive and diffusion-controlled contributions at 0.6 mV s^−1^, in which the pseudocapacitive fraction is shown in the red region. (**d**) Normalized contribution ratio of pseudocapacitive at different scan rates. (**e**) GITT voltage profiles of SnS_2_/RGO. (**f**,**g**) Discharge and charge reaction resistances. (**h**, **i**) The corresponding D_Na_ values in the first cycle.

**Figure 6 nanomaterials-11-01932-f006:**
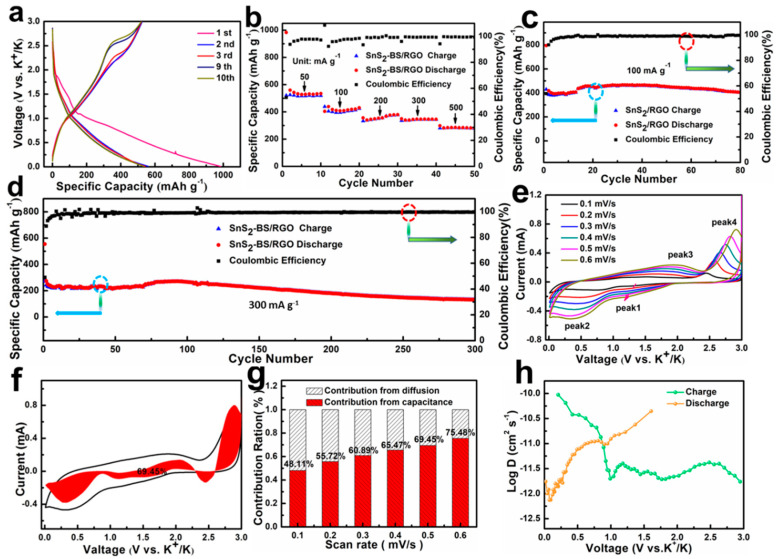
(**a**) Discharge/charge profiles of SnS_2_/RGO at 50 mA g^−1^ in the potential range of 0.01–3.0 V vs. K/K^+^. (**b**) Rate capability of SnS_2_/RGO. (**c**,**d**) Cycling performance of SnS_2_/RGO at 100 mA g^−1^ and 300 mA g^−1^, respectively. (**e**–**g**) The kinetics and quantitative analysis of the K^+^ storage behavior for SnS_2_/RGO anodes. (**h**) Calculated diffusion coefficients during charging and discharging from SnS_2_/RGO.

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
