# Peer review of "SnS_2_ Nanosheets with RGO Modification as High-Performance Anode Materials for Na-Ion and K-Ion Batteries"

_nanomaterials, 2021, doi:10.3390/nano11081932_

Round 1
Reviewer 1 Report
The manuscript "SnS2 nanosheets with RGO modification as high-performance anode materials for Na-ion and K-ion batteries" is very interesting and is well written. The abstract gives a concise summary of the manuscript. The results are also adequate and well analysed/evaluated. The conclusions highlighted and summarised the contents of the manuscript. The manuscript will fit really well within the scope of the magazine, therefore, I will recommend its acceptance as it is.
Author Response
Reply to the Reviewer: Thanks a lot for your kind comments
Reviewer 2 Report
The manuscript contains detailed studies on the SnS2 nanosheets with RGO and worth to be published in Nanomaterials. However, some modifications are required prior to publication. Please provide the surface area and pore size measurement. This can help to better illustrate the performance of the batteries. Please provide more explanations for figure 4c. In both cases, the capacity increases after some cycles and in the case of the SnS2/RGO the capacity drops in the middle of cycles and then improves again. Why? If it is possible, the authors are encouraged to provide ICP-MS of the samples. Please add some sentences regarding preparation of Na ad K anode and their optimum thickness for the testing in half cell. Do the authors believe this material can be used in supercapacitors or Li or Na/S batteries? The introduction can be expanded. The following references may help.
ChemEngineering 2020, 4(2), 42
ChemEngineering 2020, 4(3), 43
Reviewer 3 Report
I believe that the paper entitled "SnS2 nanosheets with RGO modification as high-performance anode materials for Na-ion and K-ion batteries" can be published in Nanomaterials after the authors consider the following points:
1.I believe that the coulombic efficiency=discharge/charge. In that perspective, I suggest not to calculate the initial CE and simply comment on the irreversibly as already stated.
2. What about the comparison of your work with others in the literature? Are the values similar or superior? Please comment accordingly.
3. What about the structural or morphological stability of your samples? Please comment.
4. Please explain a bit more how Figure 4f arised. It is also important to comment on the outcomes of the particular Figure. For instance, what happens with the fresh cells?
5. Further discussion is also required in Figure 5. I believe that it is not enough to simply describe the plot (For instance Figure 5e and 5h).
Overall, the paper is novel, well-written and interesting in a wide range of researchers. Nevertheless, discussion is missing to explain the figures and compare them with the literature. This part is important to strengthen the novelty of work.
Reviewer 4 Report
This manuscript presents detailed experimental results on the synthesis of SnS2 nnosheets with reduced graphene oxide modification as anode for Na and K-ions batteries.
to me the manuscript is well written and all the contents are clearly reported and sound roboust.
The method used here can be applied also to other metarials and processes and the improvement in batteries' performances is good.
I recommend the publication
Author Response

(The authors gave the same response as above.)
